# Bayesian Reasoning with Trained Neural Networks

**DOI:** 10.3390/e23060693

**Published:** 2021-05-31

**Authors:** Jakob Knollmüller, Torsten A. Enßlin

**Affiliations:** 1Physics Department, Technical University Munich, Boltzmann-Str. 2, 85748 Garching, Germany; 2Max Planck Institut for Astrophysics, Karl-Schwarzschild-Str. 1, 85748 Garching, Germany; ensslin@mpa-garching.mpg.de

**Keywords:** reasoning, generative models, uncertainty quantification, deep learning

## Abstract

We showed how to use trained neural networks to perform Bayesian reasoning in order to solve tasks outside their initial scope. Deep generative models provide prior knowledge, and classification/regression networks impose constraints. The tasks at hand were formulated as Bayesian inference problems, which we approximately solved through variational or sampling techniques. The approach built on top of already trained networks, and the addressable questions grew super-exponentially with the number of available networks. In its simplest form, the approach yielded conditional generative models. However, multiple simultaneous constraints constitute elaborate questions. We compared the approach to specifically trained generators, showed how to solve riddles, and demonstrated its compatibility with state-of-the-art architectures.

## 1. Introduction

Thanks to the recent progress in deep learning for conditional generation, we know how to build systems that can address questions such as: “How does an animal that is gray and has a trunk look?” As humans, we immediately picture an elephant. Performing the same task with neural networks requires large amounts of data that contain labels for all desired properties. Collecting the data, finding a suitable architecture, tuning all meta-parameters, and training the network on state-of-the-art compute nodes require enormous resources in terms of infrastructure and human involvement. Once the network is trained, we have a powerful and highly specialized system that can rapidly answer our original, as well as a small set of related questions.

There are numerous applications that warrant this effort, but our original question is probably not one of them, as are most conceivable questions. Nevertheless, an intelligent system should be capable of answering them through reasoning. Often, the relevant questions are not known beforehand, and they require flexible answering as they emerge. The purpose of this paper was proposing a way to design such intelligent systems.

The approach worked as follows. The context of a question (e.g., “animal” in our example) will be represented by a generative network, which translates latent variables into a member of the context (e.g., an image of a “cat”). These might be obtained from Variational Auto-Encoders (VAEs) [1] or Generative Adversarial Nets (GANs) [2], which learn to represent systems on which they are trained. The output of the generator is then passed to one or several classifier networks (identifying the color or the presence of a trunk). A likelihood distribution quantifies the adherence of the generator output with the posed constraint. Together with the prior probability distribution, this constitutes a Bayesian inference problem in terms of the latent variables of the generator. There is no need to train a specialized network for each new question, but solving the inference problem itself can still be a computationally expensive operation.

Of course, this approach relies on the availability of the individual networks as fundamental building blocks. These, however, are simple, serve a single purpose, and can be recycled throughout a large variety of questions. The scope of questions that can be answered this way grows super-exponentially with the number of available building blocks. For a set of *n* possible features $f_{i}$ of an object *o* with i∈{1,…n}, there are $m=2^{n}$ possible combinations of their presence $p\left(f_{i}|o\right)\in \left\{\mathrm{True},\mathrm{False}\right\}$. A binary question about this object can test for any of the subsets of these *m* distinguishable object realizations. The number of subsets of a set with *m* elements is $2^{m}$, as for each element, one can decide individually whether it is part of the subset or not. Thus, there are $2^{\left(2^{n}\right)}$ possible questions (page 13 of Jaynes [3]).

For example, in the example with the context animal and the two features $f_{1}=g$ for “gray” and $f_{2}=t$ for “trunk” $4=2^{2}$, animal types can be distinguished $\left\{g\wedge t,\;g\wedge \neg t,\;\neg g\wedge t,\;\neg g\wedge \neg t\right\}=:\left\{a_{1},\ldots a_{4}\right\}$, allowing for the following $16=2^{4}$ possible questions: Is it an animal with $\left\{⌀,\;a_{1},\;a_{2},\;a_{3},\;a_{4},\;a_{1}\vee a_{2},\;a_{1}\vee a_{3},\;a_{1}\vee a_{4},\;a_{2}\vee a_{3},\;a_{2}\vee a_{4},\;a_{3}\vee a_{4},\;\neg a_{1},\;\neg a_{2},\;\neg a_{3},\;\neg a_{4},a_{1}\vee a_{2},\;\vee a_{3}\vee a_{4}\right\}$?

Not all these conceivable questions are reasonable, and many are nonsense. However, just because a certain combination of traits does not occur in reality does not mean the question is invalid. With the presented approach, we can ask for things that do not exist, for example a purple elephant. This property allows exploring truly novel concepts, compared to the strict recitation of the training set of conventional deep learning approaches.

To solve a specific problem, the task has to be translated into the corresponding inference problem by assembling the individual modules, which represent the parts in the question. Missing modules might have to be acquired by a dedicated training process, but once obtained, they can be added to a library for future use.

Therefore, our approach is no replacement for traditional conditional generators. As we will demonstrate, it is better to train a dedicated network for a specific task, not factoring in the initial data curation and fine-tuning. It can work along side and on top of existing architectures. Its strength is the flexibility to answer questions the designer of the individual components did not originally have in mind. With this flexibility, it is potentially possible to approach far more elaborate tasks, for which no training data are available.

Do the answers adhere to the rules of logic? Yes, as Bayesian probabilistic reasoning is the generalization of binary logic (true or false) to uncertainties, i.e., a statement has a probability p∈[0,1] to be true [4]. Thus, binary logic is embedded in this approach and applied in the presence of certainty. Our second example demonstrates logical reasoning, as it solves a mathematical riddle in terms of hand-written digits.

Here, we provided a proof of concept and illuminated certain facets of the approach. As demonstrations of the technique, we first discuss a generator subject to one constraint, second, how to use multiple constraints to pose and solve riddles, and, third, how to combine the approach with conventional measurement data in a large-scale inference problem using modern network architectures.

## 2. Related Work

Deep generative models as a description of complex systems are especially helpful in various imaging-related problems, such as de-noising, in-painting, or super-resolution by recovering the latent variables of the generator [5,6]. Even untrained generators can provide good models for such tasks [7]. The mentioned methods rely on point-estimates that return a single solution as the answer. Using the deep generative models in a Bayesian context, the posterior distribution allows accounting for complex uncertainty structures [8,9]. In essence, we followed these works and extended them by adding constraints that were expressed through trained classification or regression networks. The fundamental idea was also proposed by Nguyen et al. [10]. Here, we rigorously derived the approach from a Bayesian perspective, showed how this can lead to reasoning on complex tasks, and explored its properties. We discussed how this could provide a path to more general intelligent systems.

Many of the tasks addressed by the methods above are also directly approached by deep learning through end-to-end solutions, for example super-resolution [11]. Similarly, our approach gave another perspective on conditional generative models. Usually, these are obtained by providing labels or more general constraints [12,13,14,15] during a training phase. We instead used unconstrained generators and flexibly added further constraints through independently trained networks in a modular fashion, allowing combining information from multiple sources. This way, we had a clear understanding of the role of the individual modules, despite they themselves remaining a black box.

Some related methods manipulate the latent variables of unconstrained generators to change samples in a desired direction [16]. Furthermore, more implicit information in the form of the preference of one sample over the other can guide the generation [17]. Our posterior distribution was in principle also a manipulation of the latent variable distribution, such that the constraints were fulfilled.

Interestingly, we can regard our approach as a form of continual variational learning [18]. When approximating our posterior variationally, we conceptually added an additional set of layers to the original generator. This was due to the reparametrization trick [1], which formulates the approximate distribution in terms of a simple source distribution together with a transformation, i.e., another generator. The new layers are then responsible for satisfying the posed constraint. Repeating this for multiple constraints, we can partially train generators with the help of already trained classification/regression networks, instead of data alone.

## 3. Trained Neural Networks as Models

### 3.1. Constraints through Neural Networks

We often know about one or several properties of some system *x*. With some appropriate function *F*, we can check whether these properties y=F(x) are actually present. In our case, such functions trained a neural network. Given a desired feature value *y* as the data, a candidate system *x* is judged for its adherence to the feature via a likelihood probability, i.e., P(y|F(x)). For continuous quantities with some given uncertainty, we often can choose the Gaussian distribution:(1)$$\begin{array}{c} P(y|F(x))=N(y|F(x),N)\;. \end{array}$$

Here, F(x) serves as the mean of the Gaussian and *N* is the covariance matrix. The more certain a feature is, the narrower the distribution is centered on its mean.

In the case of discrete categories, the function *F* might provide classification probabilities $p_{i}\left(x\right)$ for the feature of *x* being in class *i*. An appropriate choice for a likelihood could then be the categorical distribution:(2)$$\begin{array}{c} P\left(y|F\left(x\right)\right)=C\left(y|F\left(x\right)\right)=p_{y}\left(x\right)\;. \end{array}$$
Mathematically, this distribution describes the outcome of one trial. It does not directly encode how much we trust the estimate of the network. A simple way to introduce this is to raise this distribution to a certain power α.
(3)$$\begin{array}{c} P\left(y|F\left(x\right)\right)\propto C^{\alpha }\left(y|F\left(x\right)\right)=p_{y}^{\alpha }\left(x\right)\;. \end{array}$$

Positive integer values for α are equivalent to multiple consecutive draws with the same outcome, resembling the multinomial distribution. This is analogous to a narrower variance of the Gaussian in the continuous case, and we used this parameter to encode our certainty in a categorical feature.

In the case of multiple constraints, e.g., $y=(y_{1},y_{2})$, we can expand the likelihood using the product rule:(4)$$\begin{array}{c} P\left(y_{1},y_{2}|F\left(x\right)\right)=P\left(y_{1}|y_{2},F\left(x\right)\right)P\left(y_{2}|F\left(x\right)\right) \end{array}$$

Ideally, we would like to assume independence between the two constraints, as this allowed us to compile the likelihoods as independent modules. In the limiting case of certainty, i.e., a delta-likelihood, this assumption holds, as there is no conditional dependence. Therefore, in the case of small uncertainties, neglecting the dependence can be justified as well.

Otherwise, the question is where the potential dependence of the data arises. This origin determines whether the dependency needs to be explicitly modeled or whether independent likelihoods can be assumed for the different measured data features. In order to gain clarity on this question, we distinguished and discussed three different origins:intrinsic correlations of features;the measurement of similar or identical features;correlations in the noise among the different feature measurements.

Intrinsic correlations of features appear when quantities of the investigated system are coupled. For example, the age and height of children are strongly correlated. However, measured heights and ages have independent likelihoods, despite the fact that the true ages of children might affect the ability to measure their height accurately (by their differing ability to stay motionless). This is the case because the measured age has no influence on the height measurement, but only the true age does. The likelihood is the probability to obtain the observed data given the property of the investigated object. The correlation of these properties is not encoded in the likelihood, they are part of the prior.

Measurement of similar or identical features requires a bit more thought. Here, one has to distinguish between independent measurements, such as measuring the height of a child with two independent devices, each having its own noise process, and dependent measurements, just as stating the same measurement twice as an extreme case of dependency. In the former case, the likelihoods of the two measurements are independent and in the latter case not, and in the extreme example of replicated data, only one likelihood should be used. If both data replications are to be used, their correlation has to be modeled in the likelihood.

Correlations in the noise among the different feature measurements need to be taken into account. In the example of measuring the age and height of children, this could for example occur if the measurements were performed with two neural networks that shared some part of the infrastructure, which would cause correlated disturbances to both measurements. Such correlated noise needs to be identified and characterized before the application, for example by analyzing the measurement and classification errors and their correlations with well-labeled training data. If this is then accurately described in terms of the joint likelihood function for both feature measurements, its effect will properly be taken into account and be corrected in the Bayesian inference. If not, unwanted bias will enter the results.

Judging whether independence of different features can be assumed or might not be delicate, as the boundary between the generative network representing the prior and the analyzing network representing elements of the likelihood could be placed at different locations in the overall network. At this stage of this research, we did not want to give a premature answer to the general problem of dependencies in the likelihood and recommend that the individual situations be analyzed on a case-by-case basis. The generic answer will hopefully be given by future research. Here, we assumed the independency of the different data likelihoods and accepted the potential error.

### 3.2. Deep Generative Priors

We can also encode the properties of a system within a prior probability distribution x∼P(x). An equivalent representation of this distribution is a generative model x=G(ξ). These transform latent variables with simple distributions ξ∼P(ξ) into system realizations *x*. Deep generative models, such as GANs and VAEs, represent complex systems that so far have evaded an explicit mathematical formulation. They acquire their capabilities through large amounts of examples within a training set. The Bayes theorem allows combining this generative prior with any further information in the form of a likelihood $P(y^{\prime }|x)$ with data $y^{\prime }$, posing an inference problem in terms of the latent variables of the generator.
(5)$$\begin{array}{c} P\left(\xi |y^{\prime }\right)=\frac{P\left(y^{\prime }|G\left(\xi \right)\right)P\left(\xi \right)}{P(y^{\prime })}\;. \end{array}$$

The prior of the latent variables is the simple source distribution of the generator. Without loss of generality, we can assume a standard Gaussian distribution over the latent variable P(ξ)=N(ξ|0,𝟙), as this provides a convenient parametrization for inference [19,20]. In case the generator was originally trained on another source distribution, an appropriate transformation can be absorbed in the network architecture.

## 4. Bayesian Reasoning with Trained Neural Networks

We now used a deep generative model x=G(ξ), which encoded our prior knowledge on the system and fed its output into the classification or regression network F(x) to check whether the corresponding property was fulfilled. This concatenation F∘G(ξ) relates the latent variable to the data in the likelihood. The prior itself is the source distribution P(ξ)=N(ξ|0,𝟙) of the latent variables ξ.

The Bayes theorem allowed us to combine the associated probability distributions to obtain the posterior distribution over latent variables that were compatible with the constraint,
(6)$$\begin{array}{c} P\left(\xi |y\right)=\frac{P\left(y|F\circ G\left(\xi \right)\right)N\left(\xi |0,𝟙\right)}{P(y)}\;. \end{array}$$

This is a Bayesian inference problem in terms of the latent variables of the generator. Arbitrarily many constraints, either through networks or conventional measurements, can be considered by including additional likelihoods. As discussed, either independence among the individual terms has to be assured or the likelihood has to reflect the dependence structure, for example through a parametric fit on dedicated training data.

### Approximate Inference

Due to the nonlinear structure in *F* and *G*, the evidence P(y) will not be available analytically, so we had to rely on approximations to the posterior distribution. The associated approximation problem might be challenging due to the high dimensionality of the posterior and the complex shape of the posterior distribution in the latent space caused by the posed constraints in the feature space. Sampling techniques, such as Hamiltonian Monte Carlo (HMC) [21], provide samples from this posterior, but require large amounts of computational resources. This allowed us to verify the validity of our approach, and we used it in the first example.

Variational inference [22] can be much faster than HMC. The true posterior P(ξ|y) is approximated with another distribution $Q_{\varphi }\left(\xi \right)$ within a parametrized family by minimizing their Kullback–Leibler divergence [23] (maximizing the ELBO [24]) with respect to the variational parameters φ. The reparametrization trick [1] allows expressing the approximation in terms of a deterministic function $\xi =H_{\varphi }\left(\zeta \right)$ and a transformed random variable ζ that follows a simple source distribution, e.g., ζ∼N(ζ|0,𝟙). We stochastically estimated the ELBO and its gradient through samples from the approximation [25]. The deterministic reparametrization is a generative model for latent variables ξ that are compatible with the posed constraint. Therefore, the concatenation of the unconstrained generator with this reparametrization, i.e., $G\circ H_{\varphi }\left(\zeta \right)$, gives a conditional generator with the same mathematical structure as the original one. Thus, the variational inference provides an additional set of network layers on top of the original latent layer that are responsible for satisfying the additional constraints.

The accuracy of variational inference depends on the capability of the approximate distribution to capture the true posterior. Flexible approaches, such as normalizing flows [26], allow in principle for arbitrary accuracy, but are computationally expensive. For this reason, we only considered simple Gaussian approximations here. To avoid an explicit parametrization of the full covariance, we used a mean-field approach [27]. As additional method we used was Metric Gaussian Variational Inference (MGVI) for larger problems [28], which also captures correlations among all quantities implicitly.

## 5. Demonstrations

### 5.1. Conditional Digit Generation

In the first example, we wanted to illustrate how our approach yielded conditional generative models. We constrained a generator of hand-written digits to a certain label, the value of the digit. As the likelihood, we used the categorical distribution, containing the trained classification network F(x) attached to the output of the generator x=G(ξ). The prior for latent variables ξ is the source distribution of the generator, i.e., the standard Gaussian. The posterior is proportional to the product of prior and likelihood with a certain choice for α to express our confidence in the constraint.
(7)$$\begin{array}{c} P\left(\xi |y\right)\propto C^{\alpha }\left(y|F\circ G\left(\xi \right)\right)\;N\left(\xi |0,𝟙\right) \end{array}$$

As the generative model, we used a Wasserstein-GAN [29,30] with three hidden layers, a convolutional architecture, and 128 latent variables trained on the MNIST dataset [31]. The digit classification was performed by a deep three-layer convolutional neural network [32] trained on cross-entropy and achieving 98% test accuracy. We strongly enforced the constraint by setting α=100.

As a reference, we used a dedicatedly trained, conditional Wasserstein-GAN (cWGAN) [12,33]. This network used the same architecture and training routine as for the unconstrained GAN above. We provided the label information in one-hot encoding in the latent layer of the generator and in the form of an outer product with the image to the discriminator. The capabilities of the networks should therefore be comparable. We used 8000 samples for each category.

In this example, we solved the inference problem in two ways, via HMC sampling and a variational mean-field approximation. For every digit, we used eight HMC-chains to draw 80,000 samples in total, after disregarding an initial burn-in and tuning phase. We aimed for an HMC acceptance rate of 0.6 and adapted a diagonal HMC mass matrix. For every sample, we performed 10 leapfrog integration steps, and all chains were initialized at a prior sample. One chain required 36 minutes. Each chain ran on one core of an AMD EPYC 7662 64-Core Processor. To ensure convergence, we calculated the Gelman–Rubin test statistic $\hat{R}$ for all latent variables [34]. Their mean value was $\hat{R}=1.001$, which indicated well-converged chains. The average effective sample size was $N_{\mathrm{eff}}=5474$.

For the variational mean-field approximation, we employed a modified version of the Adam optimizer [35], as described in Kucukelbir et al. [20]. We chose η=1 and optimized for a total time of 600 seconds. To stochastically estimate the KL-divergence, we used five pairs of antithetic samples. Antithetic sampling [36] allows reducing the stochasticity of the estimates on gradient and loss. Every sample drawn from the approximation was accompanied by a totally anti-correlated partner, obtained by mirroring the sample at the center of the Gaussian. As for HMC, we also used 8000 samples from the approximate posterior for our further analysis.

The results for all methods and digits are illustrated in Figure 1. Visually, the HMC and cWGAN samples were almost indistinguishable. Both exhibited strong morphological variability, while satisfying the posed constraint. Compared to them, the mean-field samples were more uniform in shape, but mostly readable. As the variational approximation was unimodal and tended to underestimate the uncertainty of the posterior, it could not express all the fringe structures of the target distribution and narrow in on a typical mode.

For a more quantitative evaluation of the methods, we classified the samples using two instances of the digit classifier, the one used in the model, as well as an independently trained network with the same architecture. We compared the sample classification to the posed constraint and calculated an accuracy, which is given in Table 1.

In addition to that, we used the Fréchet Inception Distance (FID) [37] as a metric to evaluate the quality of the generated samples. It used a trained Inception-v3 architecture [38] to project an ensemble of samples through several convolutional layers into a 2048-dimensional feature space. In this space, we compared the multivariate Gaussian statistics of ensembles of different origin. As a reference, we always used the original MNIST training set with the respective label, and we compared it to the samples from the various methods. Ideally, those sets were statistically indistinguishable, in which case the FID vanished. We also report this metric in Table 1.

As indicated by the visual assessment, we found a strong similarity between cWGAN and HMC for the independently trained classifier with an overall accuracy of 96.5% and 95.6%, respectively. The cWGAN performed slightly better, which was not surprising as it was trained to directly perform this task. The mean-field approximation had a comparable accuracy of 95.8%. When using the classifier from the model, the HMC samples were almost exclusively classified correctly. This illustrated that all network properties were inherited to the inference, including their flaws. As HMC explores all posterior features, the pathological cases will be part of the samples. Interestingly, the mean-field samples appeared to be unaffected. As the approximation settled on a typical mode, the samples were less controversial and, therefore, more robust against the employed classifier.

Regarding the Fréchet inception distance, the samples from HMC actually outperformed the cWGAN results with averages of 22.3 and 28.6, respectively. The posterior samples therefore had a statistically better resemblance to the MNIST dataset. The visual impression of the more uniform samples from the mean-field approximation was reflected in the higher FID of 39.9. As expected, the approximate posterior performed worse than the full distribution. However, in some cases (for example, the digit eight), it also outperformed the cWGAN and therefore might be a valid option in many applications.

To demonstrate the convergence behavior of HMC, we show the FID of all samples during the warmup phase for all chains in Figure 2. Most chains dropped off fast, but overall, the FID was higher in this initial phase due to burn-in, as well as strongly correlated samples. Note that a portion of the observed decrease can be attributed to the increased number of samples with time, as the estimate of the statistics became better.

### 5.2. Solving Riddles

Here, we wanted to solve a riddle by enforcing multiple constraints simultaneously. We knew a priori that we were looking for three single-digit, handwritten numbers. We wanted them to fulfil the five constraints outlined on the left of Figure 3. The only viable solution was the combination 134. In the model, the three digits were generated through three instances of the same generator used in the previous example, i.e., x=G(ξ), resulting in a total of 384 latent variables in ξ. For each of the five constraints, we assembled a function $F_{i}\left(x\right)$ that checked whether it was fulfilled or not. For Constraints I, IV, and V, those corresponded to three independently trained convolutional neural networks applied to the respective digit. For the fourth constraint, we re-used the digit classification network from the previous example. The other two constraints used the same architecture, but were trained on the respective task, i.e., whether $x_{1}\in \left\{1,3,5,7,9\right\}$ or $x_{3}\in \left\{0,6,8,9\right\}$, respectively.

The remaining Constraints II and III involved multiple numbers simultaneously. Both required the classification probabilities of the digits to calculate how likely they were satisfied. For every digit, this was a 10-dimensional vector. The mathematical logics were directly implemented in the model, represented by a two-tensor *A* and a three-tensor *B* with ones at locations corresponding to valid expressions and zeros elsewhere. Contracting these tensors with the classification probabilities provided the overall probability the constraint was fulfilled. The graphical structure of the inference problem is outlined on the right of Figure 3.

The posterior distribution was proportional to the product of all likelihood terms and the prior (We note that Constraint IV contains redundant information, as the combination of I, II, and III already enforced the non-evenness of the third digit. This, however, was a property of the prior model (that used Arabic digits) and not of the way these properties were measured. Therefore, we can regard their measurements as independent and just multiply their likelihoods. In other words, the ability of one network to detect a seven (given that digit) was independent of the ability of another network to spot the respective constraints.).
(8)$$\begin{array}{c} P\left(\xi |y_{I},\ldots ,y_{V}\right)\propto \prod_{i\in I\ldots V}C^{\alpha }\left(y_{i}|F_{i}\left(G\left(\xi \right)\right)\right)\times N\left(\xi |0,𝟙\right) \end{array}$$

As we were only interested in the solution of the riddle, and not the full posterior structure, we performed a variational mean-field approximation. The challenge in this case was the multi-modality, as there were many combinations of numbers that partially fulfilled the constraints. An annealing strategy [39] partially mitigated this issue. Initially, we chose a small α to allow the optimization to explore the posterior landscape and, later on, increased it to learn the structure of the resulting mode.

Regarding the optimization scheme, we had issues with the convergence of common momentum-based stochastic optimizers and, therefore, used non-stochastic optimizers in combination with good gradient estimates, antithetic sampling, and line-search.

Five sample pairs were used to estimate the ELBO, its gradient, and the other quantities. The overall runtime was 2400 s. We chose α∈{0.5,1,3,10} and increased it after every 600 s. To illustrate the behavior quantitatively, we repeated the approximation for 100 different random seeds.

In the end, eighty-eight percent of the runs ended up with the correct solution. To track the progress during the optimization, we followed the evolution of five quantities. First, the ELBO for the final α=10 as the overall optimization goal. Second, our accuracy was in terms of the average conditional categorical likelihood that the samples showed the correct solution. Third, the average conditional categorical likelihood over samples and constraints was a score to quantify how well the conditions were met. These quantities were between zero and one. The fourth and fifth showed cumulatively the fraction of runs that were able to achieve exclusively or any correct samples up to that time. All these quantities, averaged over the hundred runs, are shown in Figure 4.

In the first quarter, the constraints were only weakly enforced, and the approximation explored the posterior distribution. For most runs, the first occurrence of a correct sample fell in this section, and afterwards, this line flattened. The capability to explore was in contradiction to well approximating the local mode, so only in a small number of cases, exclusively correct samples were achieved in this phase. At the end, the score was significantly above the accuracy, so although all constraints were satisfied, the solution itself was not necessarily correct.

How this is possible can be seen in the two examples shown in Figure 5. Here, samples and their mean from two different runs are shown. The first one correctly identified the solution, whereas the second one ended up in an almost correct mode. In this, all constraints were satisfied, except that some “cheating” was involved in fulfilling Constraint IV: the last digit was an eight, drawn in such a way that it had no closed circles in order to comply with all constraints. Qualitatively, all runs that did not show the correct solution ended up in this mode. This was possible because two distinct networks were used to check for the constraints, and these samples corresponded to a fringe case in which both were satisfied. This is reminiscent of adversarial examples [40].

To avoid this behavior, additional information could be added to the likelihood. In fact, our third constraint, i.e., the last digit not being a seven, was mathematically not necessary. Its purpose was to remove a similar local minimum to make it easier to find the correct solution. Based on this, a heuristic could be developed to iteratively add insights to wrong solutions to the problem to come up with the correct answer.

### 5.3. Reconstructing Faces

In the last example, we reconstructed faces from degraded, noisy, and incomplete data, making use of the additional information of age and gender. We compared them to reconstructions without this additional information, using only the image data. As the generative model of face images x=G(ξ), we used the stylegan architecture trained on the Flickr-Faces-HQ dataset [41]. It generated photo-realistic images of faces at a resolution of 1024×1024 pixels from 512 latent parameters and contained 23 million weights. We used the ResNet-50 architecture [42], trained on the IMDB-WIKI dataset [43,44] for age and gender estimates, $F_{a}\left(x\right)$ and $F_{g}\left(x\right)$, respectively, which contained 10 million weights each.

We compiled a set of faces drawn from the generator as the ground truth and degraded them in several ways to obtain the image data. The three color channels were added up to generate a grayscale image. Then, the resolution was reduced to 64×64 pixels via coarse-graining, followed by masking the left part. These steps are summarized in the linear degradation operator $F_{I}\left(x\right)$. Finally, Gaussian white noise with unit variance was added, providing image data $y_{I}$ and noise covariance $N_{I}=𝟙$. We obtained age and gender estimates of the ground truth by applying the corresponding classifier. To account for different input and output shapes, a re-scaling from 1024×1024 to 224×224 pixels linked the generator and classifiers.

All likelihood terms contained the generator applied to the respective operator or network. The image data entered via a Gaussian likelihood. For the age prediction, we calculated the weighted average provided by the classification probabilities, resulting in a continuous estimate. On this, we also imposed a Gaussian likelihood, centered on the true age $y_{a}$ and assuming a standard deviation of one year, i.e., $N_{a}=1$. The gender was constrained via a categorical likelihood with data $y_{g}$ in favor of the respective category, and we used an α=10. The posterior was then proportional to:(9)$$\begin{array}{cc} P(\xi |y_{I},y_{a},y_{g})\propto & \;N(y_{I}|F_{I}\left(G\left(\xi \right)\right),N_{I})\;\\ & \times N(y_{a}|F_{a}\left(G\left(\xi \right)\right),N_{a})\;\\ & \times C^{\alpha }\left(y_{g}|F_{g}\left(G\left(\xi \right)\right)\right)\;N\left(\xi |0,𝟙\right). \end{array}$$

The posterior distribution was approximated using MGVI [28] instead of the previous mean-field approach to achieve faster convergence. We performed fifteen iterations with five antithetic sample pairs and thirty natural gradient steps, followed by another five steps with twenty-five pairs. MGVI is an iterative procedure, not directly optimizing an ELBO. The hardware used in this example was an Intel Xeon CPU E5-2680 with 2.4 GHz and an NVIDIA GeForce GTX 1080ti. One reconstruction required about 30 h.

The ground truth, data, and results with and without the age and gender information for five cases are shown in Figure 6. Overall, the sample means in both cases exhibited strong similarity with the ground truth including facial expression, posture, the background, and overall color scheme. Small-scale features, such as hairstyle and background details, were washed out. The gender seemed to be already well constrained by the image data, and adding this information did not provide much benefit.

In contrast to that, the age information significantly impacted the result. In the case it was provided, the variance was visually reduced. This made sense, as the age was mostly associated with small-scale or color features, which were removed by the degradation. In Figure 7, we show the deviation of the attributed age of the samples from the ground truth over all faces. This validated our visual impression, as the deviation was small in the case the age information was provided and the samples followed the posed constraint.

Given the bad quality of the image data, the fidelity of the reconstruction might be surprising. Every image from the generator was parametrized by 512 latent numbers, which could only describe a vanishingly small subset of all possible face images. The reconstruction was only performed within this subset, resulting in this accuracy. A more realistic generator would cover a wider range of possibilities and is expected to provide a larger variety of posterior samples and, therefore, a less accurate reconstruction.

Another issue with this approach was the inherited bias from the employed networks. If a certain feature was not sufficiently constrained by any further information, the result reproduced the prior and, therefore, the balances within the training set of the generator. In our examples, this could be observed for age, as well as ethnic features. Furthermore, the classifiers might push the results in unintended directions. However, with our approach, we could explicitly add additional information through additional networks to counteract this behavior and de-bias the results to some extent.

## 6. Conclusions

We demonstrated how to conditionally sample from trained generative models with the help of trained classification/regression networks. By combining several constraints, we could build a collection of neural networks that jointly solved novel tasks through reasoning and quantifying their uncertainty. The approach could built upon state-of-the-art network architectures and provided answers in the form of potentially high-dimensional posterior distributions. Information on high-level concepts could be included to support reconstructions with conventional measurement data. The optimal configuration, architectures, and inference schemes for the reasoning need to be identified by future research.

One could envision large-scale reasoning systems that flexibly answer a large variety of complex questions by assembling appropriate modules from a library of trained networks. The inherent super-exponential reach could allow building systems that derive conclusions for changing tasks. Therefore, our approach could be a step towards more generic artificial intelligence. One has to be aware of unintended biases that enter through the trained networks. These influence the reasoning, leading to potentially unintended consequences.

Furthermore, our approach posed the question of whether similar processes are at work in human/natural intelligence. The reasoning of our approach entirely depended on information that was already stored within networks, without the need for further external input. Depending on the task, generative and discriminative networks are connected on the fly. As this backward-deduction is computational expensive, repeatedly occurring tasks might not be solved this way, but are better written into forward models. Nevertheless, any new cognitive task of this kind an intelligence faces might initially be solved in the way described here, and later optimized if required.

## Figures and Tables

**Figure 1 entropy-23-00693-f001:**
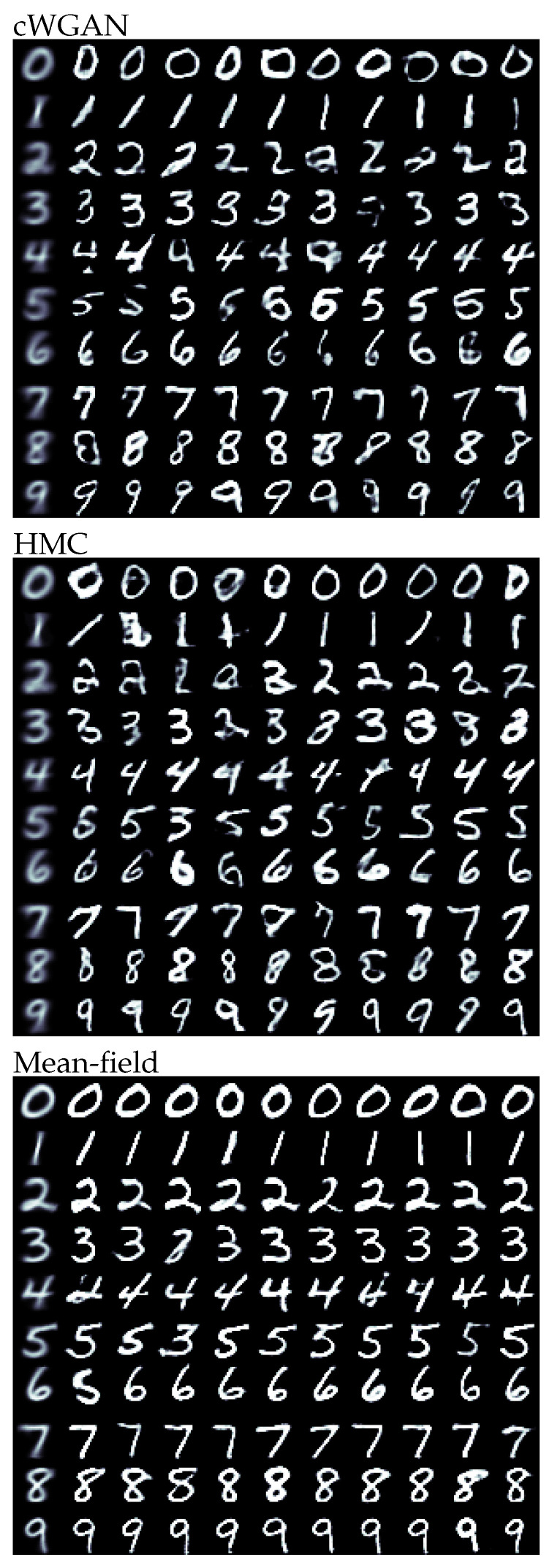
The sample mean (leftmost columns) and samples (other columns) obtained from cWGAN (**top**), HMC (**center**), and a Gaussian mean-field approximation (**bottom**) for the task to generate a handwritten digit of a certain label.

**Figure 2 entropy-23-00693-f002:**
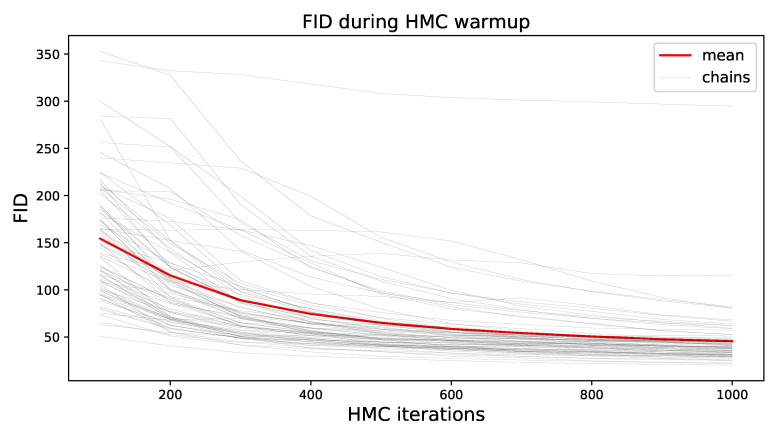
The FID for all current samples and chains during the warmup phase of HMC.

**Figure 3 entropy-23-00693-f003:**
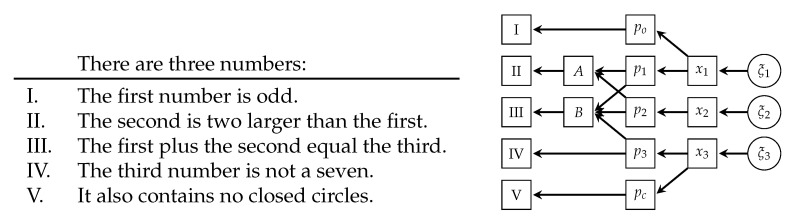
The riddle discussed in Section 5.2 (**left**) and the graphical structure of the associated riddle solver (**right**). Here, $p_{i}$ indicates the classification probability given the respective network (*o* for odd, *c* for circle, and 1, 2, and 3 for the digits).

**Figure 4 entropy-23-00693-f004:**
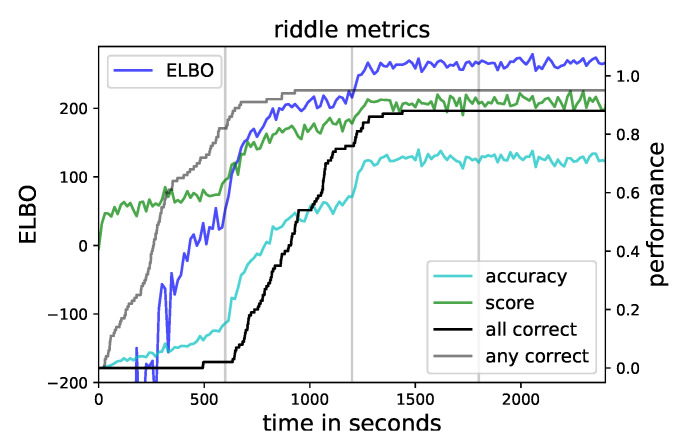
The ELBO and various other metrics of the riddle during the optimization. Vertical lines indicate the increases in α.

**Figure 5 entropy-23-00693-f005:**
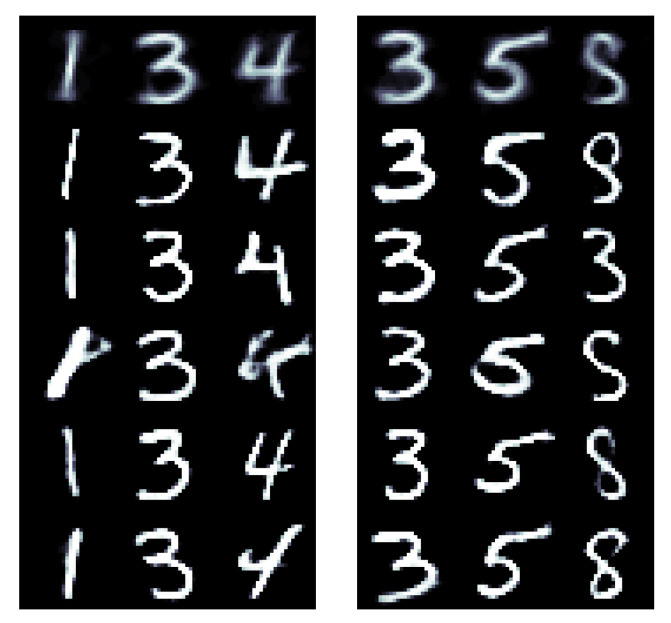
Two examples for the provided solutions for the riddle. The left one shows a correct solution, whereas the other one is only almost correct. The top rows show the ensemble means.

**Figure 6 entropy-23-00693-f006:**
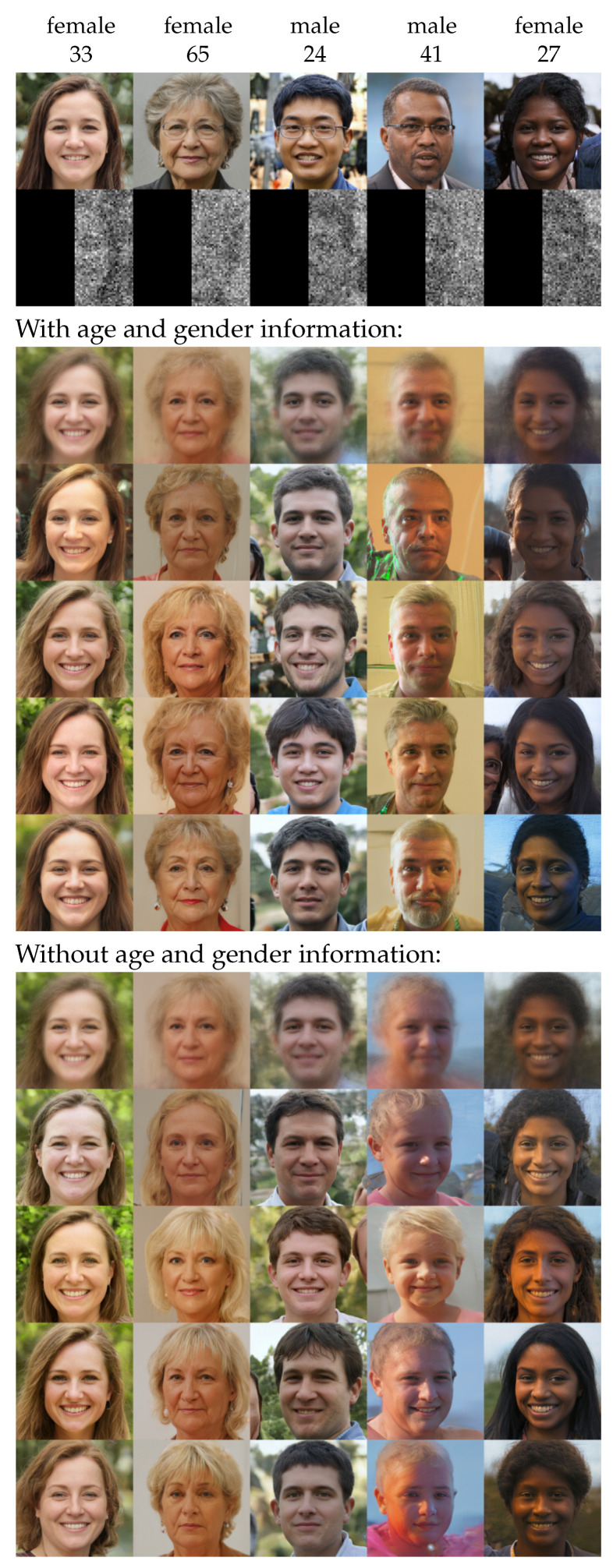
The setup and results for the face reconstructions. The top panel gives the five ground truth images with age and gender, as well as the degraded image data. The central panel shows the reconstruction for the full information. It gives the sample mean (top row) and random samples (rest). The bottom panel is analogous for a reconstruction using only the image data and no age and gender information.

**Figure 7 entropy-23-00693-f007:**
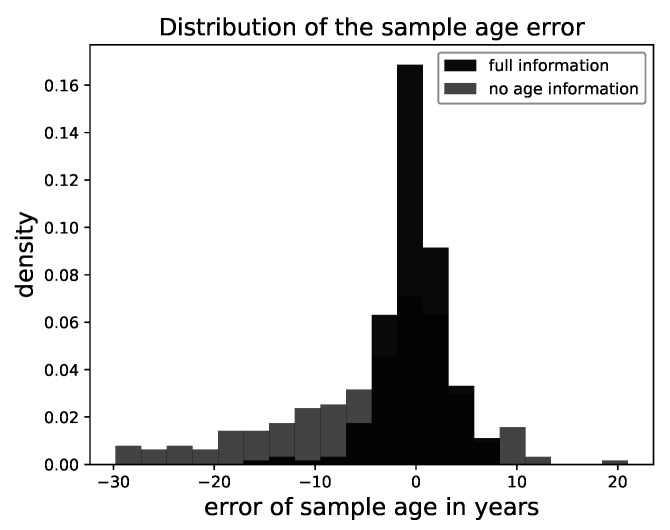
A histogram of the deviation between the sample age and true age for all cases.

**Table 1 entropy-23-00693-t001:** Classification Accuracies (ACC) and Fréchet Innovation Distance (FID) for the respective constraint and method for conditional digit generation. Accuracies without brackets are obtained using an independently trained classifier. Brackets show the result of the classifier used within the model.

Method	Metric	All	0	1	2	3	4	5	6	7	8	9
cWGAN	ACC	0.965	0.99	0.99	0.94	0.97	0.96	0.94	0.98	0.96	0.96	0.96
		(0.967)	(0.99)	(0.98)	(0.97)	(0.95)	(0.96)	(0.95)	(0.99)	(0.94)	(0.95)	(0.97)
	FID	28.6	18.0	14.7	35.9	29.2	33.8	27.3	26.8	29.8	42.8	27.8
HMC	ACC	0.956	0.99	0.96	0.89	0.97	0.96	0.94	0.96	0.98	0.96	0.96
		(0.997)	(1)	(0.98)	(1)	(1)	(1)	(1)	(0.99)	(1)	(1)	(1)
	FID	22.3	18.2	19.6	33.8	25.6	19.2	21.4	25.3	19.0	27.2	13.6
Mean-field	ACC	0.958	0.97	0.97	0.95	0.97	0.93	0.96	0.97	0.96	0.96	0.94
		(0.960)	(0.96)	(0.97)	(0.96)	(0.96)	(0.94)	(0.96)	(0.97)	(0.96)	(0.96)	(0.95)
	FID	39.9	36.1	37.7	43.8	31.7	28.2	25.6	36.5	44.4	39.4	31.6

## Data Availability

Not Applicable.

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
