# Peer review of "Bayesian Reasoning with Trained Neural Networks"

_entropy, 2021, doi:10.3390/e23060693_

Round 1

Reviewer 1 Report

This work proposes to use deep generative models and constants to perform  Bayesian reasoning. The concept is novel, and the idea is demonstrated in conditional digit generation, solving riddles, and reconstructing faces. The manuscript is well written, and the flow is very smooth. I enjoy reading this paper. I would recommend the publication of this manuscript.

Only a minor comment for curiosity, could the authors comment on whether “building a collection of neural network” would suffer from the curse of dimensionality? i.e., how does the number of neural network scales with the size of the task?

Reviewer 2 Report

Review of Remote Sensing Manuscript ID: 1203270
Bayesian Reasoning with Trained Neural Networks

This reviewer finds the paper to be aimed at solving an important problem with a novel method. This reviewer would be happy to see this paper published in Entropy. However, this reviewer has some major concerns which are elaborated below and recommends that the authors submit a major revision for re-consideration.

  1. The authors state in lines 40-42 that for n available feature-detection networks, the possible combinations is 2^n. This is technically true. However, in reality, there is only a lower dimensional manifold with admissibility constraints. For example, if we work with the authors’ example of grey animal with trunk. The possibilities are many: elephant, anteater, etc. However, if we ask for a cat with 3 ears or a dog with 5 mouths, the sample space is empty. This is despite having networks for mouth, cat, dog, and ear detection. This reviewer recommends that the authors carefully revise these sentences to talk about a practical upper bound in addition to a technical upper bound. Furthermore, would it be possible to share some of these network trunks in a multi-task-like setup since the initial layers of the neural network will be quite similar (detecting edges, blobs, etc.)?
  2. What do the authors mean when they say “model of the underlying rate” on line 119?
  3. It would be good to get the subtle but important point a little more attention: “independence in likelihood does not mean independence of the variables”. This can be done by having the two likelihood functions defined for equation (4) in terms of the parameters $\theta_1$ and $\theta_2$ of the neural networks approximating these probabilities and saying that $\theta_1$ and $\theta_2$ are independent.
  4. Question 3 brings this reviewer back to the question this reviewer asked in Question 1: would it be possible to share some of these network trunks in a multi-task-like setup? If likelihoods are assumed independent, the weights cannot be shared? Meaning, does the authors’ proposed framework work only when all feature detectors are independent?
  5. Perhaps, it would be good to expand on line 126 in a dedicated paragraph and discuss the advantages (computational, number of parameters, admissible neural architectures for feature detectors) and disadvantages (inductive biases, etc.) of the likelihood independence assumption. This is crucial to the paper and this reviewer strongly recommends a thorough discussion of this.
  6. This reviewer is not certain that VAEs and GANs model “complex systems that evade an explicit mathematical formulation”. While the neural network is a complex mathematical model representing x, it has so far not been conclusively proved that NO such model can exist. Perhaps, the authors would consider rephrasing this.
  7. Equation (5) has a subtle mathematical nuance where it is assumed that G is a deterministic function since (5) will not hold if G is a stochastic mapping of $\xi$. While GANs typically have deterministic generators, VAEs famously do not. However, this section has been motivated with GANs and VAEs. This reviewer recommends that the authors always make such assumptions explicit.
  8. How do the authors propose to “correct for their dependence” in line 137? This is quite crucial and must be exemplified. Without this, this paper seems incomplete since the case where there are independent variables is straight-forward.
  9. It seems from table 1 that cWGAN and HMC struggle with generating 2 and 5. Why is that? And why does mean-field approach solve this problem?
  10. It would also be helpful if the authors could comment on the compute time and compute resources required for the 3 approaches in table 1 so that the results can be put into context.
  11. The performance of the presented framework for the face reconstruction example is questionable. Can the authors quantify the results in a more systematic manner? This reviewer is not able to appreciate how the framework helps in conditional generation specifically when the generations in figure 5 don’t appear to be drastically different. Perhaps this reviewer is missing out some details which the authors can elaborate on in a revision.
  12. Finally, the key context required in this paper is a thorough comparison of compute time and compute resources of the proposed frameworks with existing conditional generative models. Such a comparison is strongly recommended by this reviewer.

Round 2

Reviewer 2 Report

Second Review of Remote Sensing Manuscript ID: 1203270
Bayesian Reasoning with Trained Neural Networks

The authors have attempted to answer some of the concerns raised by this reviewer. However, they have chosen to not answer two critical questions below:

  1. This reviewer is still not sure how weight sharing in a multitask framework does not induce dependence of likelihoods. In fact, the motivation of methods like MTAN and cross-stitch is to introduce such dependencies so as to enable holistic decision making. Can the authors please comment and thoroughly describe on why “multi-task setups and weight-sharing is not an issue”?
  2. This reviewer does not agree with the authors’ comment in the response: “[no] meaningful way to compare the methods without doubling the length of the paper while providing little further insight.” Simple comparisons like FID scores of the generated samples, evolution of this score as the HMC iterations increase, etc. are simple yet efficient methods to show that the model is doing something meaningful. One can also add an additional appendix page stating the hardware the authors used for the experiments, what times they observe, state their methodology of how they calculated this time, and describe a paragraph on where they clearly see many improvements possible. One may also state that this is a PoC and readers may want to take into consideration this fact before they start implementing the methods proposed. Putting this work in context would only add to the thorough analysis already present in the paper and help readers analyze literature for benchmarking.

This reviewer would appreciate if the authors added responses to the above points to the paper. This reviewer will, however, stop very close to recommending a second major revision. The paper has enough for a paper but adding responses to the above comments will make it a good paper. This reviewer therefore recommends accepting this paper for publication to Entropy and strongly recommends that the authors provide responses to the above concerns before submitting the final revision.
